# Mode Connectivity in Neural Quantum States

Vinicius Hernandes*[1], Thomas Spriggs[1], and Eliska Greplova[1]

[1]QuTech and Kavli Institute of Nanoscience, Delft, Netherlands
{v.hernandes, t.e.spriggs, e.greplova}@tudelft.nl

## Abstract

We investigate the phenomenon of mode connectivity in Neural Quantum States, a neural network-based approximation to quantum wavefunctions. By analyzing the energy landscape of networks trained to fit the ground-state of the transverse-field Ising model, we find that minima corresponding to the solutions in different physical phases exhibit distinct connectivity properties. Notably, we observe an asymmetry across the quantum phase transition: models trained in the disordered phase are linearly connected to a wider range of models than those trained in the ordered phase, suggesting a connection between the energy landscape geometry and underlying physical phenomena.

## 1   Introduction

A central challenge in computational physics is finding the ground state of many-body quantum Hamiltonians. Whilst there are efficient algorithms involving tensor networks, they face limitations for higher-dimensional systems with complex entanglement behavior [1]. An alternative is Variational Monte Carlo (VMC), where a parametrized ansatz for the wavefunction is optimized to minimize the system's energy [2]. Recently, Neural Quantum States (NQS) have emerged as a powerful VMC ansatz, using neural networks to represent quantum wavefunctions with a tractable number of parameters. An NQS maps a quantum configuration to its corresponding complex amplitude in the wavefunction [3, 4].

Despite their success, a deeper understanding of the properties of NQS is needed. Recent work has started bridging this gap by applying concepts from machine learning theory to analyze NQS, like inspecting the neural tangent kernel of trained models [5, 6], the geometry of NQS solutions [7], weight space analysis of fine tuned NQS [8], and looking for the presence of the lottery ticket hypothesis [9, 10] and double descent [11, 12]. In this work, we explore another such concept: **mode connectivity** [13–16], the observation that minima found by training can often be connected by simple, low-loss paths. We study how the connectivity of NQS solutions changes across a quantum phase transition, linking

*Corresponding Author.

the geometry of the optimization landscape to the underlying physics of the system.

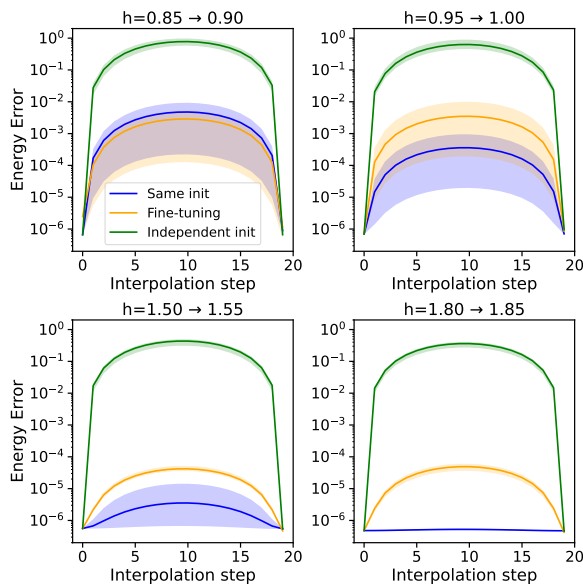

**Figure 1.** Energy error along the linear interpolation path between two trained NQS. We compare pairs of models trained for transverse-field values (0.85, 0.90), (0.95, 1.00), (1.50, 1.55), and (1.80, 1.85). We test three initialization schemes: random seeds (green), same seed (blue), and fine-tuning (yellow), with solid lines showing the mean of the energy error, and shaded area spanning from minimum to maximum values of energy error.

## 2   Methodology

We study the one-dimensional transverse-field Ising model (TFIM), a prototypical model in quantum physics described by the Hamiltonian: $H = -\sum_i \sigma_i^z \sigma_{i+1}^z - h \sum_i \sigma_i^x$. This model exhibits a quantum phase transition at a transverse field of $h = 1$, separating an ordered ferromagnetic phase ($h < 1$) from a disordered paramagnetic phase ($h > 1$).

Our NQS ansatz $\Psi_\theta$ is a neural network with parameters $\theta$. We find the ground state by variationally minimizing the energy expectation value $E(\theta) = \langle \Psi_\theta | H | \Psi_\theta \rangle / \langle \Psi_\theta | \Psi_\theta \rangle$. For the small system sizes considered here, $N = 10$ spins, we perform an exact summation over all $2^N$ configurations, avoiding Monte Carlo sampling noise.

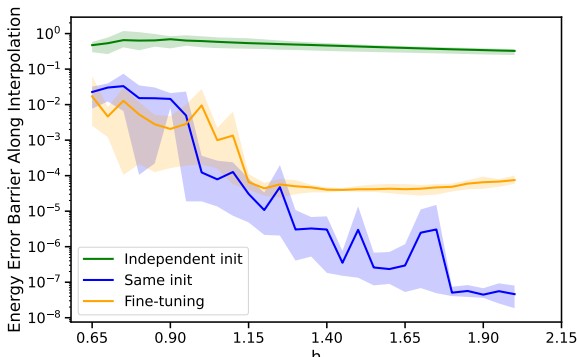

**Figure 2.** Energy error barrier height for pairs of models trained at nearby fields $(h, h + 0.05)$ across the phase diagram. The barrier is defined as the maximum error on the path minus the minimum error. For models with the same initialization (blue) and fine-tuned (yellow), the barrier is consistently lower than the randomly initialized models (green), for which the barrier is uniformly high.

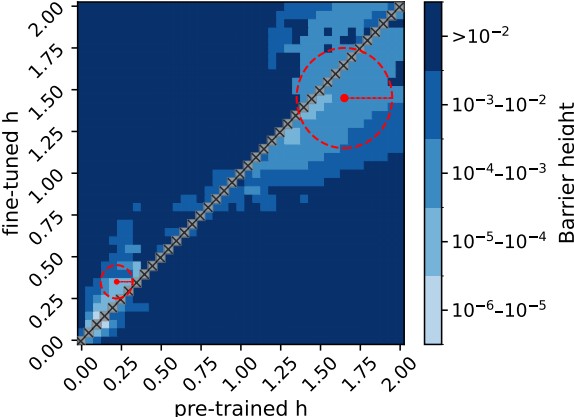

**Figure 3.** Energy error barrier height for all pairs of models $(h_{\text{baseline}}, h_{\text{fine-tune}})$. The red circles indicate a region for which a model intialized at the $h$ value at the center of the circle can be interpolated to other $h$ values while keeping the maximum energy error barrier height reasonably low.

To probe mode connectivity, we train two models with parameters $\theta_1$ and $\theta_2$ and analyze the energy along the linear interpolation path $\theta(\alpha) = (1-\alpha)\theta_1 + \alpha\theta_2$ for $\alpha \in [0,1]$. We define the **energy error barrier** as the maximum energy error along this path minus the minimum error. We perform two main experiments: (i) We train pairs of NQS for nearby fields $(h, h + 0.05)$ across the phase diagram using three strategies: initializing with different random seeds, with the same seed, and by fine-tuning one model from the other. (ii) We systematically investigate fine-tuning by training a model at a baseline $h_{\text{baseline}}$ and fine-tuning it to a target $h_{\text{fine-tune}}$ for all pairs in the range $h \in [0,2]$.

## 3 Results

**Connectivity for nearby models.** We first analyze the connectivity between NQS trained for nearby field values $(h, h + 0.05)$. Figure 1 shows four interpolation paths, while Figure 2 depicts the barrier heights across the entire phase diagram. The results reveal a clear dependence on both the initialization strategy and the physical phase. Models with different random initializations are disconnected by a uniformly high energy barrier, regardless of their position in the phase diagram. When models share initialization, we observe a distinct asymmetry around the critical point $h = 1$. The energy barrier is low but clearly non-zero in the ordered phase ($h < 1$) and drops to nearly zero in the disordered phase ($h > 1$). This suggests that while the ground state minimum evolves smoothly, the loss landscape is significantly "flatter" along interpolation paths in the disordered region. Fine-tuning one model from its neighbor also results in low barriers, which are generally higher than the same-seed case but much

lower than the random-seed case. This strategy also shows lower barriers in the disordered phase, but significantly higher than those reached with same-seed initialization.

**Fine-tuning across the phase diagram.** Our second experiment systematically explores fine-tuning between all pairs of $(h_{\text{baseline}}, h_{\text{fine-tune}})$. The results, shown in Figure 3, confirm and amplify the asymmetry seen earlier. Models pre-trained in the disordered phase ($h > 1$) maintain a low-barrier connection over a large range of $h$ values when fine-tuned. Conversely, models pre-trained in the ordered phase ($h < 1$) become disconnected by a high barrier when fine-tuned. This difference in range is shown by the red circles in Figure 3: the area of the circle for a baseline model trained in the disordered phase is larger than that of the one trained at the ordered phase. This strongly suggests that the minima in the disordered phase are geometrically "broader" and better connected to other regions of the loss landscape.

## 4 Conclusion

We showed that neural wavefunctions trained across different field values of the same Hamiltonian exhibit low energy error along the linear interpolation between solutions when initialized the same. We also show that when fine-tuning pre-trained neural wavefunctions, mode connectivity changes depending on the phase of the pre-trained model. These findings suggest that analyzing mode connectivity provides new insights into how physical phase structure shapes the optimization landscape of Neural Quantum States.

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
