# OpenReview forum: "Mode Connectivity in Neural Quantum States"
_NLDL.org/2026/Abstracts_Track — NLDL 2026 Abstracts_

### Official Review · Reviewer_HD8P · 2025-10-24

**Soundness:** 3
**Correctness:** 3
**Rating:** 4
**Confidence:** 2

**Summary:**

The authors want to leverage concepts from machine learning theory to better understand neural quantum states, particularly mode connectivity. They aim to find a neural network which minimises the energy functional, which depends on the Hamiltonian. For two solutions they investigate the error barrier, meaning the maximum error along a straight path between two solutions. They find that this barrier is lower for minima found in the "disordered phase".

**Strengths:**

- The figures seem to indicate that the ground idea is correct, clear empirical evidence.
- The problem is relevant for the ML community, as it leverages ideas from ML theory.
- The authors show the significance of the work,as it tackles problems that are interesting for the physics community. Thus it presents an interdisciplenary and novel contribution.

**Weaknesses:**

- For a ML conference, the abstract lacks insights into NQS. The problem and a lot of the important details are unclear to non-experts. The work would benefit from a more thorough explanation, to make it more interesting to Machine Learning experts.
- The sentence (032-039) about how other works apply concepts from ML Theory to NQS is irrelevant, not insightsful and confusing: these works have nothing to do with the problem at hand.
- Some statements are mathematically imprecise and too vague, for example "geometrically broader" and "better connected".
- Are there any insights which architectures should be considered? It has been shown that this choice impacts mode connectivity.

---

### Official Review · Reviewer_LVUM · 2025-10-28

**Soundness:** 3
**Correctness:** 2
**Rating:** 4
**Confidence:** 3

**Summary:**

The author analyzed the relationship between the geometry of the energy landscape and the underlying physical phenomena.
A Neural Network approximation was applied to quantum wavefunctions to determine mode connectivity values between quantum states.

**Strengths:**

The abstract “Mode Connectivity in Neural Quantum States” presents the results of the research clearly.
The proposed approach enables a controllable and interpretable exploration of the Mode Connectivity in Neural Quantum States.

**Weaknesses:**

To better align with the conference’s thematic focus, it is recommended that the title “Mode Connectivity in Neural Quantum States” be refined to more explicitly reflect the topics of the NLDL conference.

---

### Official Review · Reviewer_TxhS · 2025-10-30

**Soundness:** 3
**Correctness:** 3
**Rating:** 4
**Confidence:** 4

**Summary:**

The abstract clearly defines and explains the phenomenon observed in Neural Quantum States (NQS) when evaluating how effectively their energy landscapes can model the one-dimensional quantum system known as the Transverse Field Ising Model (TFIM). The results reveal an asymmetry in the learned representations, supporting the proposed relationship discussed in their experiments, findings, and conclusions. Overall, the work presents interesting findings that encourage further discussion and exploration, and makes for an interesting submission.

**Strengths:**

Experimental results are presented even though it is an abstract submission to indicate the connection of physical phenomena and landscape geometry in the TFIM model.

**Weaknesses:**

Perhaps they could expand on what further work they would perform to continue the investigation and build towards a paper with further experiments, results, and applications.

---

### Decision · Program_Chairs · 2025-11-05

**Decision:**

Accept

**Comment:**

The abstract is of interest to the community and should be presented at the conference.